

# Prediction of human fetal–maternal blood concentration ratio of chemicals

Chia-Chi Wang[1], Pinpin Lin[2], Che-Yu Chou[3], Shan-Shan Wang[3] and Chun-Wei Tung[2,3]

[1] Department and Graduate Institute of Veterinary Medicine, School of Veterinary Medicine, National Taiwan University, Taipei, Taiwan
[2] National Institute of Environmental Health Sciences, National Health Research Institutes, Miaoli County, Taiwan
[3] Graduate Institute of Data Science, Taipei Medical University, Taipei, Taiwan

## ABSTRACT

**Background**. The measurement of human fetal-maternal blood concentration ratio (logFM) of chemicals is critical for the risk assessment of chemical-induced developmental toxicity. While a few in vitro and ex vivo experimental methods were developed for predicting logFM of chemicals, the obtained experimental results are not able to directly predict in vivo outcomes.

**Methods**. A total of 55 chemicals with logFM values representing in vivo fetal-maternal blood ratio were divided into training and test datasets. An interpretable linear regression model was developed along with feature selection methods. Cross-validation on training dataset and prediction on independent test dataset were conducted to validate the prediction model.

**Results**. This study presents the first valid quantitative structure-activity relationship model following the Organisation for Economic Co-operation and Development (OECD) guidelines based on multiple linear regression for predicting in vivo logFM values. The autocorrelation descriptor AATSC1c and information content descriptor ZMIC1 were identified as informative features for predicting logFM. After the adjustment of the applicability domain, the developed model performs well with correlation coefficients of 0.875, 0.850 and 0.847 for model fitting, leave-one-out cross-validation and independent test, respectively. The model is expected to be useful for assessing human transplacental exposure.

## INTRODUCTION

The placenta, a barrier between fetal and maternal circulation, plays important roles in the maintenance of fetus growth and development. The blood-placenta barrier protects the fetus from exposure to pharmaceuticals and environmental pollutants (*Myllynen, Pasanen & Pelkonen, 2005*). Placental permeability of chemicals is one of the critical endpoints to evaluate the chemical-induced developmental toxicity (*Myllynen, Pasanen & Pelkonen, 2005*; *Myren et al., 2007*). Traditionally, the assessment of teratogenic and fetotoxic effects of chemicals was based on rodent models. In that way, placental transfer of chemicals can be

Corresponding author
Chun-Wei Tung,
cwtung@tmu.edu.tw

measured in intact biological systems. However, since placenta is the most species-specific organ, human cell lines and tissue models are considered more appropriate for evaluating the transfer of chemicals across the human placental barrier (*Leiser & Kaufmann, 1994*; *Myllynen, Pasanen & Pelkonen, 2005*; *Giaginis, Theocharis & Tsantili-Kakoulidou, 2012*).

Several methods were developed for evaluating the placental transfer of chemicals including in vitro assays based on primary trophoblastic cells and immortal placental human cell lines (*Vähäkangas & Myllynen, 2006*; *Myren et al., 2007*) and an ex vivo human placental perfusion model (*Miller et al., 1993*). The ex vivo human placental perfusion model preserving intact structural integrity is able to mimic maternal and fetal blood circulation and is therefore useful for studying the placental transfer of chemicals (*Miller et al., 1993*). While the in vitro and ex vivo models provide valuable tools for analyzing placental transfer of chemicals, they are both labor-intensive and time-consuming. Most importantly, the results from in vitro and ex vivo methods are not able to directly predict in vivo outcomes making the assessment of placental transfer difficult (*Hutson et al., 2011*).

In contrast, in vivo data provides the most direct evidence for the assessment of chemical toxicity. in vivo data can be obtained by measuring drug concentrations in the umbilical cord blood and maternal blood at delivery. The fetal-maternal concentration ratio is a widely used indicator of placental permeability that has been applied to drug monitoring (*Chappuy et al., 2004*; *Ripamonti et al., 2007*; *Boyce, Hackett & Ilett, 2011*). Despite the usefulness of in vivo data, ethical concerns in maternal-fetal medicine prohibit the in vivo studies for new drugs and toxic substances (*Bourget, Roulot & Fernandez, 1995*; *Fukata et al., 2005*; *Zhang et al., 2017*). in vivo data of placental transfer is therefore very scarce. The development of computational models capable of predicting in vivo placental transfer of chemicals is desirable both for providing a better understanding of placental transfer and prioritization of chemicals of toxicity concerns for further testing.

To date, three computational models have been developed to address the quantitative structure-activity relationship (QSAR) between chemical descriptors and the placental transfer of chemicals. Two out of the three models are focused on the prediction of ex vivo human placental perfusion results due to the availability of data (*Giaginis et al., 2009*; *Zhang et al., 2015*). Only one model has been developed for predicting in vivo fetal-maternal blood concentration ratio data (logFM) (*Takaku et al., 2015*). The in vivo data was manually curated from published literature of clinical studies. Three features of molecular weight (MW), topological polar surface area (TopoPSA), and maximum E-state of hydrogen atom (Hmax) were found to be correlated to logFM. While an acceptable test performance was reported with a moderate correlation coefficient of 0.714 between observed and predicted logFM values, the QSAR model is focused on the theoretical investigation of structure-activity relationship and is not ready for practical uses due to two major issues. First, its performance should be further improved to fit the widely accepted performance criterion with an $R^2$ value larger than 0.6 to be considered as a good model (*Alexander, Tropsha & Winkler, 2015*). Second, its applicability domain (AD) should be defined to be practically useful according to the Organisation for Economic Co-operation and Development (OECD) QSAR guideline (*OECD, 2007*).
This work presents the first valid QSAR model for predicting logFM values of chemicals. To provide an interpretable prediction of logFM, the QSAR model was developed based on multiple linear regression. A four-step feature selection method was utilized to identify two informative features with reasonably good performance whose correlation coefficient values are 0.796, 0.759 and 0.808 for model fitting, leave-one-out cross-validation (LOOCV) and independent test, respectively. Subsequently, a novel AD adjustment method was proposed to generate exclusion rules for identifying chemicals out of the AD based on the classification and regression tree (CART) (*Breiman, 2017*) analysis. After the application of the defined AD, the developed QSAR model achieved high performance with correlation coefficients of 0.875, 0.850 and 0.847 for model fitting, leave-one-out cross-validation (LOOCV) and independent test, respectively. The QSAR model is expected to be useful for predicting in vivo placental permeability of chemicals.

## MATERIAL AND METHODS

### Dataset

The logFM values representing in vivo fetal-maternal blood ratio of 55 chemicals were obtained from a previous work collecting in vivo data from 16 published studies (*Takaku et al., 2015*). The 55 chemicals were randomly divided into a training dataset and a test dataset with 41 and 14 chemicals, respectively. The training dataset is utilized for feature selection, cross-validation of models, and training the final model for independent tests on the test dataset. For the development of quantitative structure-activity relationship (QSAR) models, 1-dimensional (1D) and 2-dimensional (2D) descriptors including physicochemical properties were calculated for each chemical using the PaDEL-Descriptor v2.21 software (*Yap, 2011*). PaDEL-Descriptor has been shown to be useful for developing QSAR models for several endpoints (*Takaku et al., 2015*; *Huang et al., 2015*; *Tseng et al., 2017*) and currently it is able to calculate 1,875 descriptors (1,444 1D, 2D descriptors and 431 3D descriptors) and 12 types of fingerprints. In this study, a 1,444-dimensional feature vector for each chemical was generated consisting of 1,444 1D and 2D descriptors for model development. The data tables of training and test datasets are available as Tables S1 and S2, respectively.

### Model development and feature selection

To construct an interpretable model for logFM prediction, a four-step feature selection method was developed to identify the most important feature set for developing multiple linear regression models. The first three steps of the feature selection are to remove features with (1) extreme values that are at least 100-fold larger than average, (2) more than or equal to 50% zero values (scarcity), and (3) small variation (less than 8 unique values). In the fourth step, the Lasso method (*Tibshirani, 1996*) was applied to select informative features from the remaining feature set based on a leave-one-out cross-validation (LOOCV). Subsequently, Z-score (*Kreyszig, 1979*) was applied to normalize the informative features for training multiple linear regression models as shown in Eq. (1).

$$z = (x - \mu)/\sigma, \tag{1}$$

where z, x, μ and $\sigma$ represent the normalized feature value, original feature value, mean of the feature and variance of the feature, respectively. Since features with scarcity and small variation are less informative descriptors that are not useful for modeling chemicals, these features were excluded from subsequent analysis. In this study, the LassoCV and LinearRegression functions of the scikit-learn package (*Pedregosa et al., 2011*) were utilized to implement the Lasso-based feature selection and model training, respectively.

## Applicability domain

To determine the applicability domain (AD) of the developed models, a decision tree-based method was proposed to identify the structural descriptors of chemicals prone to be incorrectly predicted. We conducted a rigorous method (*Tung, Lin & Wang, 2019*) to adjust the AD of the developed models solely based on the training dataset and evaluate the AD based on the test dataset. A four-step algorithm was described as follows. First, the absolute error for each chemical was calculated based on the LOOCV on the training dataset. Second, the feature vectors and absolute errors of chemicals were analyzed by a classification and regression tree (CART) (*Breiman, 2017*) to identify the structural rules with the highest absolute errors. The DecisionTreeRegressor function of scikit-learn package was utilized to conduct the decision tree analysis. Third, the rule with the highest absolute error was recursively identified and applied to exclude chemicals out of the AD. Finally, the adjusted AD was applied to identify chemicals in the test dataset within the AD for calculating the performance of the developed models.

## RESULTS AND DISCUSSION

### Model development

To develop a prediction model for logFM, a four-step feature selection method was applied to identify informative features. The 1,444 features were firstly analyzed following the first three feature selection steps. A total of 18, 507 and 22 features with extreme values, scarcity and small variation, respectively, were excluded from the subsequent study. The Lasso feature selection was then applied to identify two informative features of AATSC1c and ZMIC1 representing the average centered Broto-Moreau autocorrelation (lag 1, weighted by charges) and Z-modified information content index (neighborhood symmetry of 1-order), respectively, from the remaining 897 features based on the training dataset.

The two features were utilized to train and cross-validate a model based on the training dataset. The fitting results of logFM for chemicals in the training dataset is shown in Table 1 and the leave-one-out cross-validation (LOOCV) results are shown in Fig. 1. Please note that the feature values shown in Table 1 are z-score normalized values. Their corresponding mean and variance are 1.32E-04 and 8.35E-06 for AATSC1c and 4.65E+01 and 3.33E+02 for ZMIC1, respectively. The developed model is shown in the following Eq. (2).

$$\text{logFM} = -0.0882 \times \text{AATSC1c} - 0.2139 \times \text{ZMIC1} - 0.3161. \tag{2}$$

The negative coefficients in Eq. (2) indicate a negative correlation between the two features and the logFM, i.e., an increase of the two features of AATSC1c and ZMIC1 result in a decrease of the logFM value. To further clarify the importance of the features,
**Table 1  Training dataset with normalized features, logFM values and applicability domain (AD) information.**

| Name | AATSC1c | ZMIC1 | Observed logFM | Predicted logFM | AD |
|---|---|---|---|---|---|
| Oxychlordane | 0.90 | 1.90 | −1.02 | −0.80 | Y |
| DDE | 1.06 | 1.10 | −0.98 | −0.64 | Y |
| Mifepristone | 1.10 | −0.03 | −0.96 | −0.41 | N |
| Atazanavir | −0.36 | 1.94 | −0.89 | −0.70 | Y |
| Nonachlor | 1.54 | 2.54 | −0.84 | −1.00 | Y |
| Chlordane | 1.46 | 1.88 | −0.78 | −0.85 | Y |
| HCB | 1.69 | 1.23 | −0.70 | −0.73 | Y |
| Flupenthixol | 0.58 | 0.70 | −0.62 | −0.52 | Y |
| Lopinavir | 0.11 | 1.76 | −0.62 | −0.70 | Y |
| Propranolol | 0.57 | −0.34 | −0.59 | −0.29 | Y |
| Disopyramide | 0.86 | 0.17 | −0.59 | −0.43 | Y |
| Piperacillin | −1.09 | 0.76 | −0.57 | −0.38 | Y |
| Heptachlor epoxide | 1.25 | 1.41 | −0.48 | −0.73 | Y |
| Etidocaine | 0.85 | −0.59 | −0.47 | −0.26 | Y |
| Buprenorphine | 0.77 | 0.33 | −0.46 | −0.45 | Y |
| Oxprenolol | 0.25 | −0.68 | −0.43 | −0.19 | Y |
| Didanosine | −1.41 | −1.18 | −0.42 | 0.06 | N |
| Norbuprenorphine | 0.47 | −0.09 | −0.31 | −0.34 | Y |
| Clindamycin | −0.27 | −0.12 | −0.30 | −0.27 | Y |
| Lidocaine | 0.73 | −0.83 | −0.26 | −0.20 | Y |
| Clonazepam | 0.57 | −0.16 | −0.23 | −0.33 | Y |
| Flecainide | −0.96 | 0.47 | −0.20 | −0.33 | Y |
| Nevirapine | 0.07 | −0.77 | −0.17 | −0.16 | Y |
| Remifentanil | 0.03 | 0.07 | −0.14 | −0.33 | Y |
| Ethabutol | −0.06 | −0.86 | −0.12 | −0.13 | Y |
| Nifedipine | −0.07 | −0.31 | −0.11 | −0.24 | Y |
| Acebutolol | 0.08 | −0.61 | −0.10 | −0.19 | Y |
| Clonidine | −0.42 | −0.72 | −0.05 | −0.13 | Y |
| Ticarcillin | −1.67 | −0.20 | −0.04 | −0.13 | Y |
| Lamivudine | −2.50 | −1.24 | −0.03 | 0.17 | Y |
| Chlorpyrifos | −1.46 | 0.04 | −0.01 | −0.20 | Y |
| Indomethacin | −0.33 | −0.13 | −0.01 | −0.26 | Y |
| Metronidazole | −0.49 | −1.39 | 0.00 | 0.03 | Y |
| Diazinon | −0.73 | −0.51 | 0.00 | −0.14 | Y |
| Metoprolol | 0.54 | −0.74 | 0.00 | −0.20 | Y |
| Abacavir | −0.97 | −0.71 | 0.01 | −0.08 | Y |
| Procainamide | 0.35 | −0.62 | 0.04 | −0.21 | Y |
| Zidovudine | −1.45 | −1.07 | 0.09 | 0.04 | Y |
| Diazepam | 0.79 | −0.12 | 0.10 | −0.36 | N |
| Stavudine | −2.09 | −1.13 | 0.12 | 0.11 | Y |
| Valproic acid | −0.27 | −1.15 | 0.18 | −0.05 | Y |

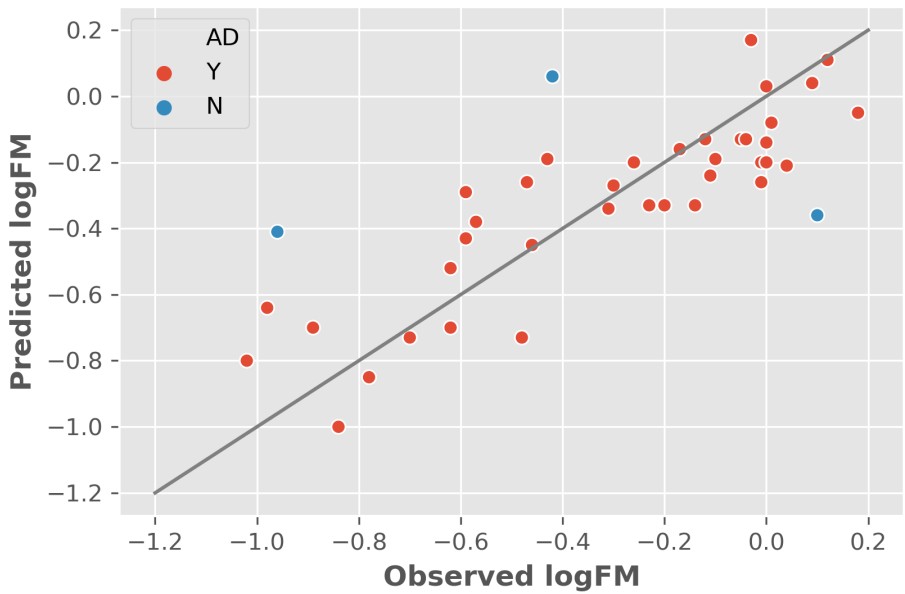

**Figure 1** **The leave-one-out cross-validation results based on the training dataset and two informative features.** Abbreviations: AD, applicability domain; Y, the chemical is in the AD (red dot); N, the chemical is out of the AD.

standard partial regression coefficients were calculated based on a piecewiseSEM package (*Lefcheck, 2016*). The partial regression coefficients for AATSC1c and ZMIC1 are −0.2586 and −0.6297 suggesting that a full shift in AATSC1c and ZMIC1 would result in a shift of 26% and 63% along the range of logFM. The correlation coefficient values of model fitting and LOOCV on the training dataset were 0.796 and 0.759, respectively. The small difference between the correlation coefficients of model fitting and LOOCV indicates a small chance of overfitting problems that is consistent with a similar correlation coefficient value of 0.808 obtained from the independent test on the test dataset. Detailed prediction results on the test dataset are shown in Table 2 and Fig. 2 presents the plot of observed and predicted logFM values. Please note that the feature values shown in Table 2 are z-score normalized values.

The y-randomization test, a widely used method for assessing the quality of a developed model by comparing the model performance with random models (*Rücker, Rücker & Meringer, 2007*), was also utilized to test the model. A total of 100 runs of LOOCV were conducted based on 100 modified training datasets whose corresponding logFM values were randomly shuffled. The mean and standard deviation of the y-randomization test are −0.202 and 0.288, respectively. The y-randomization performance is much lower than the original model (0.759) showing the uniqueness of our model.

## Adjustment of applicability domain
While the model is with acceptable performance, it is extremely important to determine the applicability domain (AD) as the number of chemicals in the training dataset is relatively small compared to the huge chemical space. By properly adjusting the AD, the chemicals

**Table 2  Test dataset with normalized features, logFM values and applicability domain (AD) information.**

| Name | AATSC1c | ZMIC1 | Observed logFM | Predicted logFM | AD |
|---|---|---|---|---|---|
| Indinavir | 0.44 | 1.54 | −1.10 | −0.68 | Y |
| Duloxetine | 1.13 | 0.04 | −0.92 | −0.43 | N |
| 17-Hydroxyprogesterone caproate | 0.90 | 0.71 | −0.70 | −0.55 | Y |
| Nelfinavir | 0.34 | 0.82 | −0.60 | −0.52 | Y |
| Bupivacaine | 0.91 | −0.38 | −0.52 | −0.31 | Y |
| Cefoperazone | −1.48 | 0.65 | −0.46 | −0.32 | Y |
| Naloxone | 0.15 | −0.75 | −0.30 | −0.17 | Y |
| Isoniazid | −0.31 | −1.35 | −0.21 | 0.00 | Y |
| Midazolam | 0.80 | −0.22 | −0.13 | −0.34 | N |
| Phthalimide | −0.80 | −0.85 | −0.09 | −0.06 | Y |
| Chloroquine | 1.24 | −0.39 | −0.03 | −0.34 | Y |
| Sotalol | −0.09 | −0.71 | 0.00 | −0.16 | Y |
| Dicloran | 0.24 | −0.62 | 0.03 | −0.21 | Y |
| Carnitine | −0.28 | −1.27 | 0.11 | −0.02 | Y |

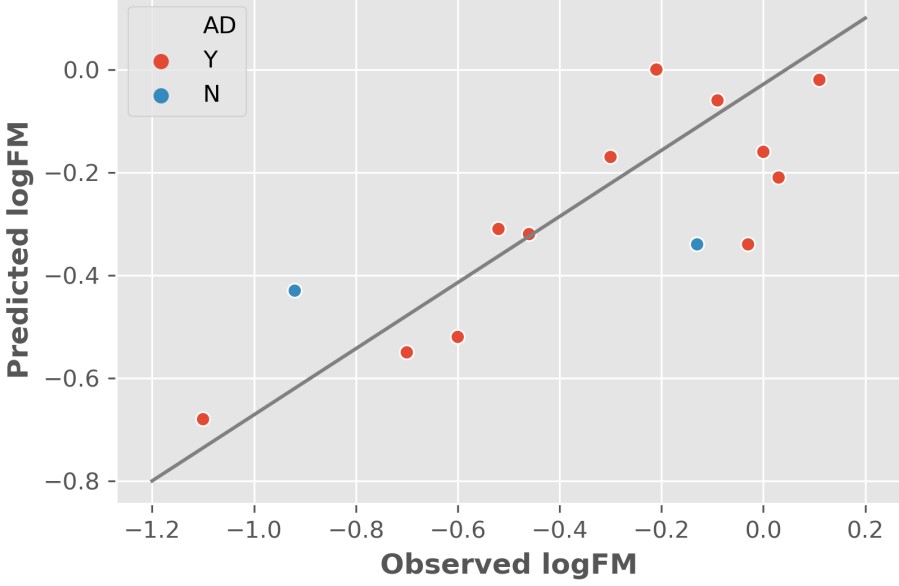

**Figure 2  The independent test results based on the test dataset and two informative features.** Abbreviations: AD, applicability domain; Y, the chemical is in the AD (red dot); N, the chemical is out of the AD.

within the defined AD of the developed model can be identified and the prediction performance is expected to be improved (*Tung, Wang & Wang, 2018*; *Tung, Lin & Wang, 2019*). To accurately evaluate the usefulness of AD for predicting unseen chemicals, the AD of the developed model was adjusted by using only the training dataset and independently tested by using the test dataset.

**Table 3 Exclusion rules for identifying chemicals out of the defined applicability domain.**

| Exclusion rule | AATSC1c | ZMIC1 |
|---|---|---|
| #1 | $0.782 < x$ | $-0.077 < x <= 0.073$ |
| #2 | $-1.957 < x <= -0.838$ | $-1.315 < x <= -1.141$ |
| #3 | $0.782 < x$ | $-0.359 < x <= -0.077$ |

In this study, we proposed a novel AD adjustment method based on the CART algorithm to identify exclusion rules for removing chemicals out of the AD. While the model performance can be continuously improved by appending more exclusion rules, a decreased coverage of chemicals within the AD can limit the practical use of the model. To balance the tradeoff between performance and coverage, we applied a stopping criterion for determining the optimal number of exclusion rules as follows. The exclusion rules were iteratively selected until no significant improvement (<1%) on the correlation coefficient was obtained by an additional exclusion rule. A total of three exclusion rules (Table 3) were selected to exclude chemicals out of AD. The exclusion rules shown in Table 3 indicate that chemicals with a relatively large AATSC1c value ($0.782 < x$) and medium ZMIC1 values ($-0.359 < x <= 0.073$) are out of AD. In addition, chemicals with small values of AATSC1c ($-1.957 < x <= -0.838$) and ZMIC1 ($-1.315 < x <= -1.141$) are out of AD. Please note that the exclusion rules are based on normalized feature values.

After the adjustment of AD, three corresponding chemicals in the training dataset were identified to be out of AD as shown in Table 1. Mifepristone, didanosine, and diazepam were excluded based on rules of #1, #2, and #3 (Table 3), respectively. The model performance was largely improved with correlation coefficient values of 0.875 and 0.850 for model fitting and LOOCV on the training dataset, respectively. The coverage of chemicals within AD is 92.68% (38/41). The defined AD was subsequently applied to exclude chemicals in the test dataset. As shown in Table 2, two chemicals were identified to be out of AD. Duloxetine and midazolam were excluded based on rules of #1 and #3, respectively. The test performance was substantially improved with a correlation coefficient of 0.847 and coverage of 85.71% (12/14) on the test dataset.

## Analysis of informative features

Two features of AATSC1c and ZMIC1 were identified as informative features. AATSC1c belongs to the autocorrelation descriptor representing the average centered Broto-Moreau autocorrelation of lag 1 weighted by charges (*Todeschini & Consonni, 2009*). The spatial charge descriptor is relevant to lipophilicity that is considered an important factor for placental transfer (*Pacifici & Nottoli, 1995*). ZMIC1 is an information content descriptor representing the Z-modified information content index (neighborhood symmetry of 1-order) (*Todeschini & Consonni, 2009*). ZMIC1 is correlated with the molecular branching, size and ring closure and size is another important factor for placental transfer (*Pacifici & Nottoli, 1995*). The selected features of AATSC1c and ZMIC1 are only moderately correlated with a correlation coefficient of 0.449 showing no multicollinearity problems. To compare with the three features (MW, Hmax and TopoPSA) identified by a previous study (*Takaku et al., 2015*), the correlation coefficients among the two informative features

**Table 4  The correlation among features identified in this study and a previous study (*Takaku et al., 2015*).**

| Correlation coefficient | MW | Hmax | TopoPSA |
|---|---|---|---|
| AATSC1c | 0.107 | −0.611 | −0.645 |
| ZMIC1 | 0.778 | −0.199 | −0.013 |

of this study and previously identified three features were calculated as shown in Table 4. The analysis showed that ZMIC1 positively correlates to MW (molecular weight) with a high correlation coefficient of 0.778. In contrast, AATSC1c negatively correlates to Hmax and TopoPSA with moderate correlation coefficients of −0.611 and −0.645, respectively.

The test performance of the proposed model using the two informative features (correlation coefficient = 0.808) is much better than the previous study using three features (0.714). As a strong correlation (correlation coefficient = 0.613) between Hmax and TopoPSA and a moderate correlation (correlation coefficient = 0.494) between MW and TopoPSA were observed, multicollinearity might be responsible for the worse performance of the previous study. The three features of MW, Hmax and TopoPSA were all excluded by the fourth step of feature selection using the Lasso method and LOOCV. The proposed algorithm coped well with the multicollinearity problem, compared with the previous study based on Akaike Information Criterion (AIC) and the whole training set. After the AD adjustment, the performance was further improved to 0.847. The two informative features are considered more relevant to logFM values.

## CONCLUSIONS

The development of computational models for predicting fetal-maternal blood concentration ratio of chemicals in humans can help the design of safer drugs and avoid unwanted toxicity by exposure to chemicals. This study presents the first valid QSAR model following the OECD principles with good prediction performance under the defined AD. The five principles are (1) a defined endpoint, (2) an unambiguous algorithm, (3) a defined domain of applicability, (4) appropriate measures of goodness-of-fit, robustness and predictivity, and (5) a mechanistic interpretation, if possible (*OECD, 2007*). The QSAR model is a multiple linear regression model based on two features of AATSC1c and ZMIC1 selected by the Lasso method. The AD of the QSAR model was determined by a novel decision tree-based analysis method and gave high correlation coefficients of 0.875, 0.850 and 0.847 for model fitting, leave-one-out cross-validation and independent test, respectively. The QSAR model is expected to be useful for predicting logFM values in humans that is an important endpoint for assessing the developmental toxicity of chemicals. Future work could be the integration of the transplacental prediction model to the weight-of-evidence framework (*Tung et al., 2020*) as one evidence. The combination of multiple in silico models could further improve the overall accuracy for prioritizing chemicals of developmental toxicity.

### Funding

This work was supported by the Ministry of Science and Technology of Taiwan (MOST-107-2221-E-038-020-MY3, MOST-107-2320-B-002-065) and National Health Research Institutes (NHRI-109A1-EMCO-0319204). The funders had no role in study design, data collection and analysis, decision to publish, or preparation of the manuscript.

### Grant Disclosures

The following grant information was disclosed by the authors:
Ministry of Science and Technology of Taiwan: MOST-107-2221-E-038-020-MY3, MOST-107-2320-B-002-065.
National Health Research Institutes: NHRI-109A1-EMCO-0319204.

### Competing Interests

The authors declare there are no competing interests.

### Author Contributions

- Chia-Chi Wang and Chun-Wei Tung conceived and designed the experiments, analyzed the data, prepared figures and/or tables, authored or reviewed drafts of the paper, and approved the final draft.
- Pinpin Lin conceived and designed the experiments, analyzed the data, authored or reviewed drafts of the paper, and approved the final draft.
- Che-Yu Chou performed the experiments, analyzed the data, prepared figures and/or tables, and approved the final draft.
- Shan-Shan Wang performed the experiments, prepared figures and/or tables, and approved the final draft.

### Data Availability

The dataset and models are available as Tables 1, 2, S1 and S2 and Eq. 1 and 2.

### Supplemental Information

Supplemental information for this article can be found online at http://dx.doi.org/10.7717/peerj.9562#supplemental-information.

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
