# Peer review of "Prediction of human fetal–maternal blood concentration ratio of chemicals"

_PeerJ, doi:10.7717/peerj.9562_

## Round 0.1 · original submission · Minor Revisions

Both reviewers recognize that your manuscript adds more meaningful data to the field. Please improve the manuscript by addressing all concerns raised by the two reviewers, especially their concerns about the experimental design.

Reviewer 1 ·

Basic reporting

The authors have reported to develop a first valid quantitative structure-activity relationship (QSAR) model following OECD guidelines based on multiple linear regression to improve prediction of in vivo human fetal-maternal blood concentration ratio (logFM) of chemicals.
This study is overall well designed and the manuscript is well written.There are a few items that need to be addressed in order to further improve the quality of the paper.

1. All the abbreviations such as OECD, AATSC1c should be defined at the first instance of using them.
2. line 107-108, please explain the biological reason the rule of the feature selection: 2) more than or equal to 50% zero values (scarcity), and 3) small variation (less than 8 unique values).
3. Please give the reference of Eq. 1.
4. line 146, Eq. 2. should be interpreted with the biological reason such as what the meaning of negative-valued in the prediction model.
5. line 156, background of y-randomization test should be given to explain why using it to test the model?
6. line 175-179, the authors should explain the 3 exclusion rules were selected to exclude chemicals out of AD. For example, 3 corresponding chemicals (Mifepristone, Didanosine, Diazepam) were identified to be out in training dataset while 3 compounds (chloroquine, didanosine, and DDE) were considered outliers because they have more than two of the standardized residuals (Takaku, T. etc. 2015).

Experimental design

To clarify the importance of the features used in multiple linear regression analysis, the standard partial regression coefficients should be calculated.

Validity of the findings

Any limitations to use this model needs to be discussed.

Additional comments

no comment

Reviewer 2 ·

Basic reporting

The manuscript is well written in general.
A few suggestions:
1) cite original data source of the in vivo fetal-maternal blood ratio of 55 chemicals.
2) Add legends to figures (e.g. color label, Y/N meaning).
3) Include a supplementary table for all the original 1444 descriptors.

Experimental design

1) Correlation coefficients of descriptors should be checked to avoid the multicollinearity before feature selection.
2) It's not clear if the authors did Feature Scaling? if so, which method used? Feature Scaling is important for linear regression. With feature scaling, the first 3 steps of feature selection may not necessary to apply.

Validity of the findings

1) Could the authors give some explanations/discussions on why the two descriptors (AATSC1c and ZMIC1) are informative, any biological significance behind them?
2) Since the current study used the same dataset and algorithm for predicting the same biological question (DOI: 10.1248/bpb.b14-00883), the author should discuss more about how and why the predicting performance improved.
3) Previous research identified MW, Hmax and TopoPSA as the best descriptors. Did these three descriptors identified as well in the original 1444 descriptor set? why they are not been picked up after feature selection?

---

## Round 0.2 · accepted · Accept

I read the revised manuscript and the rebuttal letter, and found that all reviewers' concerns have been addressed.